# Learning Deformable Tetrahedral Meshes for 3D Reconstruction

**Jun Gao**[1,2,3]        **Wenzheng Chen**[1,2,3]        **Tommy Xiang**[1,2]

**Alec Jacobson**[2]        **Morgan McGuire**[1]        **Sanja Fidler**[1,2,3]

NVIDIA[1]        University of Toronto[2]        Vector Institute[3]

{jung, wenzchen, txiang, mcguire, sfidler}@nvidia.com, jacobson@cs.toronto.edu

## Abstract

3D shape representations that accommodate learning-based 3D reconstruction are an open problem in machine learning and computer graphics. Previous work on neural 3D reconstruction demonstrated benefits, but also limitations, of point cloud, voxel, surface mesh, and implicit function representations. We introduce *Deformable Tetrahedral Meshes* (DEFTET) as a particular parameterization that utilizes volumetric tetrahedral meshes for the reconstruction problem. Unlike existing volumetric approaches, DEFTET optimizes for both vertex placement and occupancy, and is differentiable with respect to standard 3D reconstruction loss functions. It is thus simultaneously high-precision, volumetric, and amenable to learning-based neural architectures. We show that it can represent arbitrary, complex topology, is both memory and computationally efficient, and can produce high-fidelity reconstructions with a significantly smaller grid size than alternative volumetric approaches. The predicted surfaces are also inherently defined as tetrahedral meshes, thus do not require post-processing. We demonstrate that DEFTET matches or exceeds both the quality of the previous best approaches and the performance of the fastest ones. Our approach obtains high-quality tetrahedral meshes computed directly from noisy point clouds, and is the first to showcase high-quality 3D tet-mesh results using only a single image as input. Our project webpage: https://nv-tlabs.github.io/DefTet/.

## 1   Introduction

High-quality 3D reconstruction is crucial to many applications in robotics, simulation, and VR/AR. The input to a reconstruction method may be one or more images, or point clouds from depth-sensing cameras, and is often noisy or imperfectly calibrated. The output is a 3D shape in some format, which must include reasonable reconstruction results even for areas that are unrepresented in the input due to viewpoint or occlusion. Because this problem is naturally ambiguous if not ill-posed, data-driven approaches that utilize deep learning have led to recent advances on solving it, including impressive results for multiview reconstruction [35, 31], single-image reconstruction with 3D [51, 34] and 2D supervision [5, 29, 21], point clouds [41, 26], and generative modeling [1, 52, 25].

The 3D geometric representation used by a neural network strongly influences the solution space. Prior representations used with deep learning for 3D reconstruction fall into several categories: points [11, 52], voxels [8, 31], surface meshes [48, 5, 40, 29], and implicit functions (e.g., signed-distance functions) [34, 51]. Each of these representations have certain advantages and disadvantages. Voxel representations support differing topologies and enable the use of 3D convolutional neural networks (CNNs), one of the most mature 3D neural architectures. However, for high-quality 3D reconstruction, voxels require high-resolution grids that are memory intensive, or hierarchical structures like octtrees [42] that are cumbersome to implement in learning-based approaches because their structure is inherently discontinuous as the tree structure changes with occupancy.

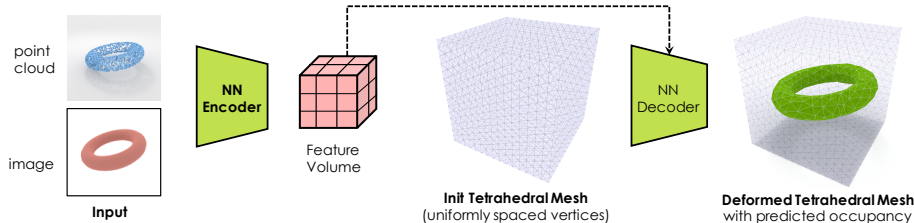

Figure 1: **DEFTET**: Given an input (either a point cloud or an image), DEFTET utilizes a neural network to deform vertices of an initial tetrahedron mesh and to predict the occupancy (green) for each tetrahedron.

Representations based on implicit functions or point clouds have the advantage that they do not impose a predetermined resolution, but they require post-processing steps to obtain a mesh [34, 32, 23, 22]. Furthermore, unlike in mesh representations, surface-based regularizations are non-trivial to incorporate in these approaches as surfaces are implicitly defined [2]. Mesh representations, which directly produce meshed objects, have been shown to produce high quality results [48, 40, 20, 5, 29]. However, they typically require the genus of the mesh to be predefined (e.g., equivalent to a sphere), which limits the types of objects this approach can handle.

Tetrahedral meshes are one of the dominant representations for volumetric solids in graphics [10, 44, 17], and particularly widely used for physics-based simulation [45]. Similar to voxels, they model not only the surface, but the entire interior of a shape using the 3D-simplex of a tetrahedron comprising four triangles as the atom. In this work, we propose utilizing tetrahedral meshes for the reconstruction problem by making them amenable to deep learning. Our representation is a cube fully subdivided into tetrahedrons, where the 3D shape is volumetric and embedded within the tetrahedral mesh. The number of vertices and their adjacency in this mesh is fixed, and proportional to the network's output size. However, the positions of the vertices and the binary occupancy of each tetrahedron is dynamic. This allows the mesh to deform to more accurately adapt to object geometry than a corresponding voxel representation. We name this data structure a *Deformable Tetrahedral Mesh* (DEFTET). Given typical reconstruction loss functions, we show how to propagate gradients back to vertex positions as well as tetrahedron occupancy analytically.

DEFTET offers several advantages over prior work. It can output shapes with arbitrary topology, using the occupancy of the tetrahedrons to differentiate the interior from the exterior of the object. DEFTET can also represent local geometric details by deforming the triangular faces in these tetrahedrons to better align with object surface, thus achieving high fidelity reconstruction at a significantly lower memory footprint than existing volumetric approaches. Furthermore, our representation is also computationally efficient as it produces tetrahedral meshes directly, without requiring post-processing during inference. Our approach supports a variety of downstream tasks such as single and multi-view image-based 3D reconstruction with either 3D or 2D supervision. Our result is the first to produce tetrahedral meshes directly from noisy point clouds and images, a problem that is notoriously hard in graphics even for surface meshes. We also showcase DEFTET on tetrahedral meshing in the wild, i.e. tetrahedralizing the interior of a given watertight surface, highlighting the advantages of our approach.

## 2 Related Works

### 2.1 3D Reconstruction

We broadly categorize previous learning-based 3D reconstruction works based on the 3D representation it uses: voxel, point cloud, surface mesh, or implicit function. We focus our discussion on voxel and mesh-based methods since they are the most related to our work.

**Voxels** store inside/outside volumetric occupancy on a regular grid. Since voxels have regular topology, voxel-based methods typically employ mature 3D CNNs yielding strong performance [47, 50, 8]. The main limitation of voxel representations is that the reconstruction quality critically depends on the grid resolution, so there is a strict tradeoff between memory consumption and fidelity. This can be seen in the difficulty of representing thin features and curves. By extending voxels to a deformable tetrahedral grid, we allow resolution to be effectively utilized. Neural Volumes [31] previously demonstrated deformable volumes as an affine warping to obtain an implicit irregular grid structure. We instead explicitly place every vertex to produce a more flexible representation. Hierarchical structures are another benefit of voxels [49, 42]. We hypothesize that our representation could also benefit from having a hierarchical structure, but we leave this to future work.

**Mesh** reconstruction work seeks to deform a reference mesh template to the target shape [48, 20, 5, 29, 53]. These methods output a mesh directly and can inherently preserve the property that the surface is as well-formed, but they are limited in their ability to generalize to objects with different topologies. New variations [15, 40, 37] that are able to modify the template's topology can, however, lead to holes and other inconsistencies in the surface because they do not model the volume of the shape itself. For example, AtlasNet uses multiple parametric surface elements to represent an object; while it yields results with different topologies, the output may require post-processing to convert into a closed mesh. [40] aims to represent objects with holes by deleting edges in the sphere topology, which can also lead to degenerated meshes. Mesh R-CNN [14] and [43] combine voxel and mesh representations by voxelizing predicted occupancy, forming a mesh, and then predicting updated positions of mesh vertices. The two steps are trained independently due to the non-differentiable operations employed in the conversion between representations. In contrast, in DEFTET, the occupancy and deformation are optimized jointly, thus being able to inform each other during training. BSP-Net [6] recursively subdivides the space using BSP-Trees. While recovering sharp edges, it cannot produce concave shapes. The most relevant work to ours is Differentiable Marching Cubes (DMC) [26], which deforms vertices in a voxel grid. However, DMC implicitly supervises the "occupancy" and deformation through the expectation over possible topologies, while our DEFTET explicitly defines the occupancy for tetrahedrons and deforms vertices for surface alignment. More importantly, the vertices in DMC are constrained to lie on the regular grid edges, which leads to many corner cases on available topologies. In contrast, we learn to deform the 3D vertices and only require that the tetrahedrons do not flip, providing more flexibility in representing geometric details.

Our work builds upon the idea of deforming a regular (2D) grid geometry to better represent geometric structures in images presented in [13], and extends it to 3D geometric reasoning and 3D applications.

## 2.2 Tetrahedral Meshing

Tetrahedral meshing is a problem widely studied in graphics. It aims to tetrahedralize the interior of a given watertight surface [10, 44, 17], supporting many downstream tasks in graphics, physics-based simulation and engineering [3, 7, 39]. One line of work, first tetrahedralizes the interior of a regular lattice and improves the boundary cells using physics-inspired simulation [36] or by snapping vertices and cutting faces [24]. QuarTet [10] further improves this approach by detecting and handling feature curves. A second line of work utilizes Delaunay-based tetrahedralization. A popular method TetGen [44] enforces inclusion of input faces in the mesh. Recent work TetWild [17] allows the surface to change within user-specified bounds, which greatly reduces unnecessary over-refinement due to surface irregularities. Most of the existing methods involve many discrete non-differentiable operations, and are thus not suitable for end-to-end deep learning.

# 3 Formulation

In this section, we describe our DEFTET formulation. We show applications of our representation to 3D reconstruction using either 3D or 2D supervision in Section 4.

## 3.1 Tetrahedron Parameterization

Without loss of generality, we assume that each 3D shape lies inside a bounded space that can then be normalized to unit cube[1]. DEFTET defines a spatial graph operating in this space, where we fully tetrahedralize the space. Each tetrahedron is a polyhedron composed of four triangular faces, six straight edges, and four vertices[2]. Two neighboring tetrahedrons share a face, three edges and three vertices. Each vertex has a 3D location, defined in the coordinate system of the bounding space. Each triangular face represents one surface segment, while each tetrahedron represents a subvolume and is expected to be non-degenerated. We use QuarTet[3] [10] to produce the initial tetrahedralization of the space, i.e., regular subdivision of the space. This initialization is fixed. Visualization is in Fig. 2.

We formulate our approach as predicting offsets for each of the vertices of the initial tetrahedrons and predicting the occupancy of each tetrahedron. Occupancy is used to identify the tetrahedrons that lie inside or outside the shape. A face that belongs to two tetrahedrons with different occupancy labels defines the object's surface. This guarantees that the output triangle mesh is "solid" in the terminology of [54]. Vertex deformation, which is the crux of our idea, aims to better align the surface faces with the ground truth object's surfaces. Note that when the deformation is fixed, our

DEFTET is conceptually equivalent to a voxel-grid, and can thus represent arbitrary object topology. However, by deforming the vertices, we can bypass the aliasing problem (i.e., "staircase pattern") in the discretized volume and accommodate for finer geometry in a memory and computationally efficient manner. Visualization is provided in Fig. 1 and 2. We describe our DEFTET in detail next.

**Tetrahedron Representation:** We denote a vertex of a tetrahedron as $v_i = [x_i, y_i, z_i]^T$, where $i \in \{1, \cdots, N\}$ and $N$ the number of all vertices. We use $\mathbf{v}$ to denote all vertices. Each triangular face of the tetrahedron is denoted with its three vertices as: $f_s = [v_{a_s}, v_{b_s}, v_{c_s}]$, with $s \in \{1, \cdots, F\}$ indexing the faces and $F$ the total number of faces. Each tetrahedron is represented with its four vertices: $T_k = [v_{a_k}, v_{b_k}, v_{c_k}, v_{d_k}]$, with $k \in \{1, \cdots, K\}$ and $K$ the total number of tetrahedrons.

We denote the (binary) occupancy of a tetrahedron $T_k$ as $O_k$. Note that occupancy depends on the positions of the vertices, which in our case can deform. We thus use a notation:

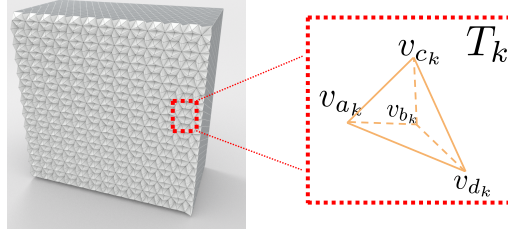

$$O_k = O_k(e(T_k)), \tag{1}$$

to emphasize that $O_k$ depends on the four of the tetrahedron's vertices. Note that different definitions of occupancy are possible, indicated with a function $e$. For example, one can define the occupancy with respect to the tetrahedron's centroid

Figure 2: Tetrahedral Mesh

based on whether the centroid lies inside or outside the object. We discuss choices of $e$ in Section 4. To not overclutter the notation with indices, we simplify notation to $O_k(\mathbf{v}_k)$ indicating that $O_k$ depends on the (correctly indexed) tetrahedron vertices.

The probability of whether a triangular face $f_s$ defines a surface of an object is then equal to:

$$P_s(\mathbf{v}) = O_{s_1}(\mathbf{v}_{s_1})(1 - O_{s_2}(\mathbf{v}_{s_2})) + (1 - O_{s_1}(\mathbf{v}_{s_1}))O_{s_2}(\mathbf{v}_{s_2}), \tag{2}$$

where $T_{s_1}$ and $T_{s_2}$ are two neighboring tetrahedrons that share the face $f_s$. This definition computes the probability of one of the neighboring tetrahedrons to be occupied and the other not.

## 3.2 DEFTET

Given input observations $I$, which can be an image or a point cloud, we aim to utilize a neural network $h$ to predict the relative offset for each vertex and the occupancy of each tetrahedron:

$$\left\{\{\Delta x_i, \Delta y_i, \Delta z_i\}_{i=1}^N, \{O_k\}_{k=1}^K\right\} = h(\mathbf{v}, I; \theta), \tag{3}$$

where $\theta$ parameterizes neural network $h$. The deformed vertices are thus:

$$v_i' = [x_i + \Delta x_i, y_i + \Delta y_i, z_i + \Delta z_i]^T. \tag{4}$$

We discuss neural architectures for $h$ in Section 4, and here describe our method conceptually. We first discuss our model through inference, i.e., how we obtain a tetrahedral mesh from the network's prediction. We then introduce differentiable loss functions for training our DEFTET.

**Inference:** Inference in our model is performed by feeding the provided input $I$ into the trained neural network to obtain the deformed tetrahedrons with their occupancies. We simply threshold the tetrahedrons whose occupancies are greater than a threshold, which can be optimized for using the validation set. Other methods can be further explored. To obtain the surface mesh, we can discard the internal faces by simply checking whether the faces are shared by two occupied tetrahedrons.

**Differentiable Loss Function:** We now introduce a differentiable loss function to train $h$ such that best 3D reconstruction is achieved. In particular, we define our loss function as a sum of several loss terms that help in achieving good reconstruction fidelity while regularizing the output shape:

$$L_{\text{all}}(\theta) = \lambda_{\text{recon}} L_{\text{recon}} + \lambda_{\text{vol}} L_{\text{vol}} + \lambda_{\text{lap}} L_{\text{lap}} + \lambda_{\text{sm}} L_{\text{sm}} + \lambda_{\text{del}} L_{\text{del}} + \lambda_{\text{amips}} L_{\text{amips}} \tag{5}$$

Here, $L_{\text{recon}}$ is a reconstruction loss that tries to align surface faces with ground-truth faces. We discuss two different reconstruction loss functions in the following subsections, depending on whether 3D supervision is available, or only 2D image supervision is desired. Similar to mesh-based approaches [48, 5], we utilize several regularizations to encourage geometric priors. We utilize a Laplacian loss, $L_{\text{lap}}$, to penalize the change in relative positions of neighboring vertices. A Delta

Loss, $L_{\text{del}}$, penalizes large deformation for each of the vertices. An EquiVolume loss, $L_{\text{vol}}$, penalizes differences in volume for each tetrahedron, while AMIPS Loss [12], $L_{\text{amips}}$, penalizes the distortion of each tetrahedron. Smoothness loss [5], $L_{\text{sm}}$, penalizes the difference in normals for neighboring faces when 3D ground-truth is available. Lambdas are hyperparameters that weight the influence of each of the terms. Details about these regularizers is provided in the Appendix. Note that $L_{\text{vol}}$ and $L_{\text{amips}}$ are employed to avoid flipped tetrahedrons.

### 3.2.1 Reconstruction Loss with 3D Ground-Truth

**Occupancy:** We supervise occupancy prediction from $h$ with Binary Cross Entropy:

$$L_{\text{occ}}(\theta) = \sum_{k=1}^{K} \hat{O}_k \log(O_k) + (1 - \hat{O}_k) \log(1 - O_k), \tag{6}$$

where $\hat{O}_k$ is the ground truth occupancy for tetrahedron $T_k$. We obtain $\hat{O}_k$ using the winding number [33, 19]. Specifically, for $T_k$, we first obtain its geometric centroid by averaging its four vertices, and calculate its winding number with respect to the ground truth boundary mesh. If the winding number of the centroid of an tetrahedron is positive, we consider the tetrahedron to be occupied, otherwise non-occupied. Note that the occupancy of a tetrahedron is a piece-wise linear function of its four vertices, and thus, is dynamically changing when the tetrahedron is deforming.

**Deformation:** Using ground truth tetrahedron occupancy, we obtain the triangular faces whose $P_s(\mathbf{v})$ equals to 1 using Equation 2. We denote these faces as $\mathcal{F}$. We deform $\mathcal{F}$ to align with the target object's surface by minimizing the distances between the predicted and target surface. Specifically, we first sample a set of points $\mathcal{S}$ from the surface of the target object. For each sampled point, we minimize its distance to the closest face in $\mathcal{F}$. We also sample a set of points $\mathcal{S}_{\mathcal{F}}$ from $\mathcal{F}$, and minimize their distances to the their closest points in $\mathcal{S}$:

$$L_{\text{surf}}(\theta) = \sum_{p \in \mathcal{S}} \min_{f \in \mathcal{F}} \text{dist}_f(p, f) + \sum_{q \in \mathcal{S}_{\mathcal{F}}} \min_{p \in \mathcal{S}} \text{dist}_p(p, q), \tag{7}$$

where $\text{dist}_f(p, f)$ is an L2 distance from point $p$ to face $f$, and $\text{dist}_p(p, q)$ is an L2 distance from point $p$ to $q$. The reconstruction loss function is:

$$L_{\text{recon3D}}(\theta) = L_{\text{occ}} + \lambda_{\text{surf}} L_{\text{surf}}, \tag{8}$$

where $\lambda_{\text{surf}}$ is a hyperparameter. Note that the loss function is differentiable with respect to vertex positions and tetrahedron occupancy.

### 3.2.2 Reconstruction Loss via Differentiable Rendering

To reconstruct 3D shape when only 2D images are available, we design a novel differentiable rendering algorithm to project the deformable tetrahedron to a 2D image using the provided camera. Apart from geometry (vertex positions), we also support per-vertex colors.

**Differentiable Renderer:** Our DEFTET can be treated as a mesh with a set of vertices $\mathbf{v}$ and faces $\mathbf{f}$. To render an RGB image with an annotated object mask, we first assign each vertex $v_i$ with an RGB attribute $C_i$ and a visibility attribute $D_i$. For each pixel $I_j$, we shoot a ray that eminates from the camera origin, passes through the pixel $I_j$ and hits some of the triangular faces in $\mathbf{f}$. We assume that $\{f_{l_1}, \cdots, f_{l_L}\}$ are hit with $L$ the total number, and that the hit faces are sorted in the increasing order of depth along the ray. We render all of these faces and softly combine them via visibility.

For each face $f_{l_k} = (v_a, v_b, v_c)$, its attribute in the pixel $I_j$ is obtained via barycentric interpolation:

$$\{w_a, w_b, w_c\} = \text{Barycentric}(v_a, v_b, v_c, I_j)$$
$$D_j^k = w_a D_a + w_b D_b + w_c D_c; \quad C_j^k = w_a C_a + w_b C_b + w_c C_c \tag{9}$$

Inspired by volume rendering [35, 46], we propose that the visibility $m_k$ of face $f_{l_k}$ in pixel $I_j$ depends on its own visibility and all the faces in front of it. Namely, if all of its front faces are not visible, then this face is visible:

$$m_k = \prod_{i=1}^{k-1} (1 - D_j^i) D_j^k \tag{10}$$

Finally, we use a weighted sum to obtain the color $R_j$ and visibility $M_j$ for pixel $I_j$ as below:

$$M_j = \sum_{k=1}^{L} m_k; \quad R_j = \sum_{k=1}^{L} m_k C_j^k \tag{11}$$

Note that the gradients from $M_j$ and $R_j$ can be naturally back-propogated to not only color and visibility attributes, but also vertex positions via barycentric weights based on chain rule.

**Occupancy:** We empirically found that predicting per-vertex occupancy and using the max over these as the tetrahedron's occupancy works well, i.e., $O_k = \max(D_{a_k}, D_{b_k}, D_{c_k}, D_{d_k})$.

**2D Loss:** We use L1 distance between the rendered and GT (input) image to supervise the network:

$$L_{\text{recon2D}} = \sum_j \left| R_j^{GT} - R_j \right|_1 + \lambda_{\text{mask}} \left| M_j^{GT} - M_j \right|_1 \tag{12}$$

Here, $M^{GT}$ is the mask for the object in the image. Note that the use of mask is optional and its use depends on its availability in the downstream application.

## 4 Applications

We now discuss several applications of our DEFTET. For each application, we adopt a state-of-the-art neural architecture for the task. Since our method produces mesh outputs, we found that instead of the Laplacian regularization loss term during training, using Laplace smoothing [16] as a differentiable layer as part of the neural architecture, produces more visually pleasing results in some applications. In particular, we add this layer on top of the network's output. We propagate gradients to $h$ through the Laplacian layer during training, and thus $h$ implicitly trains to be well synchronized with this layer, achieving the same accuracy quantitatively but improved qualitative results. Details about all architectures are provided in the Appendix.

### 4.1 Point Cloud 3D Reconstruction

We show application of our DEFTET on reconstructing 3D shapes from noisy point clouds. We adopt PointVoxel CNN [41, 30] to encode the input point cloud into a voxelized feature map. For each tetrahedron vertex, we compute its feature using trilinear interpolation using the vertex's position, and apply Graph-Convolutional-Network [27] to predict the deformation offset. For each tetrahedron, we use MLP to predict its centroid occupancy, whose feature is also obtained via trilinear interpolation using the centroid position.

### 4.2 Single Image 3D Reconstruction

Our DEFTET can also reconstruct 3D shapes from a single image. We adopt the network architecture proposed in DVR [38] and train it using either 3D supervision or 2D supervision.

### 4.3 Tetrahedral Meshing

We use this task to evaluate our approach against non-learning, non-generic methods designed solely for tetrahedral meshing. Our DEFTET can be applied in two ways: optimization-based and learning-based. In both cases, we exploit ground truth manifold surface and 3D supervision.

**Optimization-based:** We iteratively run two steps: 1) obtain the occupancy for each tetrahedron, 2) deform the tetrahedron by minimizing the $L_{\text{all}}$ loss (with $L_{\text{occ}}$ being zero) by back-propagating gradients to vertex positions using gradient descent.

**Learning-based:** To speedup the optimization, we perform amortized inference using a trained network. We first train a neural network proposed in Sec. 4.1 to learn to predict deformation offset, where we take clean point cloud (sampled points from the surface) as input. During inference, we first run prediction to get the initial deformation, and further run optimization to refine the prediction.

### 4.4 Novel View Synthesis and Multi-view 3D Reconstruction

DEFTET can also reconstruct 3D scenes from multiple views and synthesize novel views. Similarly to [35], we optimize DEFTET with a multi-view consistency loss, and propagate the gradients through our differentiable renderer (Sec. 3.2.2) to the positions and colors of each vertex. To better represent the complexity of a given scene, similar to [28], we do adaptive subdivision. To be fair with existing work, we do not use a mask (Eq (12)) in this application. Note that DEFTET's benefit is that it outputs an explicit (tet mesh) representation which can directly be used in downstream tasks such as physics-based simulation, or be edited by artists. More details are in Appendix.

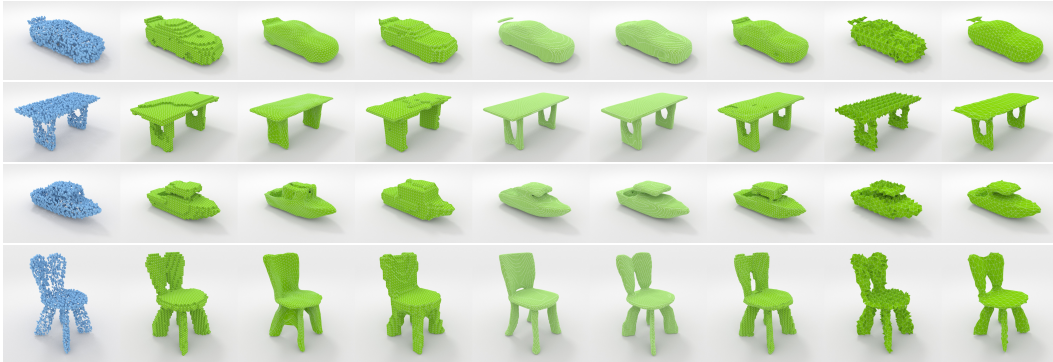

| | Input PC | 3DR2N2 [8] | Pix2Mesh [48] | DMC [26] | OccNet [34]* | OccNet [34] | MeshRCNN [14] | FIXEDTET | DEFTET |

Figure 3: Qualitative results for **Point cloud 3D reconstruction**. * denotes original OccNet [34] architecture.

| Category | Airplane | Bench | Dresser | Car | Chair | Display | Lamp | Speaker | Rifle | Sofa | Table | Phone | Vessel | **Mean-IoU** | **Time**(ms)↓ |
|---|---|---|---|---|---|---|---|---|---|---|---|---|---|---|---|
| 3D-R2N2 [8] | 59.47 | 49.50 | 78.87 | 80.80 | 63.59 | 64.04 | 42.57 | 82.12 | 52.11 | 77.77 | 56.67 | 72.24 | 66.73 | 65.11 | 174.49 |
| DeepMCube [26] | 44.72 | 39.90 | 74.33 | 76.39 | 51.01 | 56.24 | 36.25 | 55.88 | 41.80 | 70.52 | 47.19 | 74.94 | 51.64 | 55.45 | 349.83 |
| Pixel2mesh [48] | 76.57 | 60.24 | 81.96 | 87.15 | 67.76 | 75.87 | 48.90 | 86.01 | 68.55 | 85.17 | 59.49 | 88.31 | 76.92 | 74.07 | **30.31** |
| OccNet [34] | 60.28 | 47.52 | 76.93 | 78.29 | 54.51 | 66.40 | 37.18 | 77.88 | 46.91 | 75.26 | 58.02 | 77.62 | 60.09 | 62.84 | 728.36 |
| OccNet w Our Arch | 78.11 | 67.08 | 87.45 | 89.84 | 75.85 | 76.18 | 49.15 | 86.99 | 64.67 | 87.49 | 70.91 | 88.54 | 77.77 | **76.93** | 866.54 |
| MeshRCNN [14] | 71.80 | 60.20 | 86.16 | 88.32 | 70.13 | 73.50 | 45.52 | 80.40 | 60.11 | 84.01 | 60.82 | 86.04 | 70.53 | 72.12 | 228.46 |
| FIXEDTET | 67.64 | 55.48 | 81.77 | 84.50 | 65.70 | 69.29 | 45.88 | 83.03 | 54.14 | 80.96 | 58.32 | 80.05 | 69.93 | 68.98 | 43.52 |
| DEFTET | 75.12 | 66.89 | 86.97 | 89.83 | 75.20 | 79.65 | 48.86 | 87.28 | 51.99 | 88.04 | 74.93 | 91.67 | 76.07 | **76.35** | 61.39 |

Table 1: **Point cloud reconstruction** (3D IoU). DEFTET is 14x faster than OccNet with similar accuracy. All baselines but the 4th row (OccNet) are originally not designed for pc reconstruction, and thus we use our encoder and their decoder for a fair comparison. OccNet also benefits from our encoder's architecture (5th row).

# 5 Experiments

To demonstrate the effectiveness of DEFTET, we use the ShapeNet core dataset [4]. We train one model on all 13 categories. For each shape in ShapeNet, we use Kaolin [18] to preprocess and generate a watertight boundary mesh. Further details are provided in the Appendix.

## 5.1 Point Cloud 3D Reconstruction

**Experimental Setting:** We compare DEFTET with voxel [8], mesh [48], and implicit function representations [34]. We additionally compare with Deep Marching Cubes [26] and Mesh R-CNN [14]. For a fair comparison, we apply the same point cloud encoder to all methods, and use the decoder architecture provided by the authors. For OccNet [34], we additionally compare with its original point cloud encoder. Implementation of all methods is based on the official codebase[4] in [34]. We follow [48, 34, 41] and report 3D Intersection-over-Union (IoU). We also evaluate inference time of all the methods on the same Nvidia V100 GPU. Details and performance in terms of other metrics are provided in Appendix.

**Results:** Quantitative results are presented in Table 1, with a few qualitative examples shown in Fig 3. Compared to mesh-based representations, we can generate shapes with different topologies, which helps in achieving a significantly higher performance. Compared to implicit function representations, we achieve similar performance but run 14x faster as no post-processing is required. Comparing the original point cloud encoder from OccNet [34], our encoder significantly improves the performance in terms of the accuracy. We also compare to our own version of the model in which we only predict occupancies but do not learn to deform the vertices. We refer to this model as FIXEDTET. Our full model is significantly better. We also compare DEFTET with voxel-based representations at different resolutions when only trained with Chair category in Fig. 4. We achieve similar performance at a significantly lower memory footprint: our model at $30^3$ grid size is comparable to voxels at $60^3$.

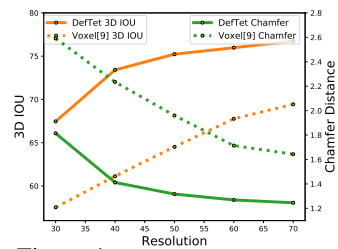

Figure 4: Point cloud reconstruction for different resolutions: voxels vs DEFTET

## 5.2 Single Image 3D Reconstruction

We are able to incorporate both 3D and 2D supervision in our approach. We focus on 2D supervision in the main paper, and provide 3D supervision results in the Appendix.

[4] https://github.com/autonomousvision/occupancy_networks

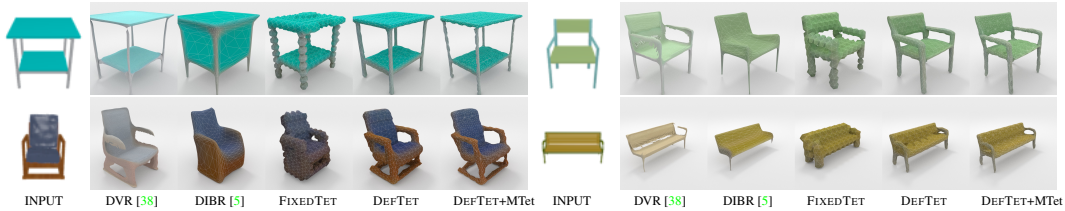

Figure 5: Examples of **Single Image Reconstruction** with **2D supervision**. Please zoom in to see details.

| Category | Airplane | Bench | Dresser | Car | Chair | Display | Lamp | Speaker | Rifle | Sofa | Table | Phone | Vessel | **Mean** | **Time**(ms)↓ |
|---|---|---|---|---|---|---|---|---|---|---|---|---|---|---|---|
| DIB-R [5] | 1.9 | 2.6 | 2.9 | 2.8 | 3.9 | 3.3 | 4.2 | 3.8 | 1.7 | 3.4 | 3.2 | 2.1 | 2.7 | 3.0 | 13.41 |
| DVR [38] | 2.2 | 2.6 | 3.0 | 2.6 | 3.3 | 3.0 | 4.4 | 3.8 | 2.0 | 3.0 | 3.5 | 2.2 | 2.8 | 3.0 | 13725.02 |
| FIXEDTET | 4.3 | 4.6 | 4.1 | 3.7 | 4.9 | 5.1 | 5.3 | 4.7 | 5.1 | 4.4 | 4.8 | 3.8 | 4.2 | 4.5 | 51.39 |
| FIXEDTET + MTet | 2.8 | 3.3 | 3.5 | 3.0 | 4.0 | 4.1 | 4.4 | 4.3 | 3.2 | 3.5 | 3.7 | 2.5 | 3.0 | 3.5 | 67.99 |
| DEFTET | 3.2 | 3.6 | 3.4 | 3.0 | 3.8 | 4.0 | 4.5 | 4.1 | 3.4 | 3.5 | 3.5 | 2.9 | 3.7 | 3.6 | 52.84 |
| DEFTET + MTet | 2.5 | 3.0 | 3.0 | 2.7 | 3.5 | 3.6 | 4.1 | 3.8 | 2.6 | 3.1 | 3.1 | 2.5 | 3.2 | 3.1 | 69.74 |

Table 2: **Single Image 3D Reconstruction** with **2D supervision**. We report Chamfer distance (lower is better).

**Experimental Setting:** In this experiment, we compare DEFTET with DIB-R [5] (mesh-based representation), DVR [38] (implicit function-based representation) and FIXEDTET, which is conceptually equivalent to a voxel representation. We adopt the same encoder architecture for all baselines as in DVR [38], but adopt their respective decoders. We train one model on all 13 ShapeNet categories. We use the same dataset as in DVR [38], and evaluate Chamfer distance and inference time for all these methods. We find that by applying Marching Tet [9] (denoted as MTet in the table) on top of our prediction allows us to extract a more accurate iso surface; details are provided in Appendix.

**Results:** Results are shown in Table 2 and Fig 5. Voxel representation (FIXEDTET) produces bumpy shapes and achieves lower scores. The mesh-based method (DIB-R [5]) runs at fastest speed since it directly outputs a surface mesh, while ours needs a few additional operations to obtain a surface mesh from the volumetric prediction. This step can be further optimized. However, due to using a fixed genus, DIB-R cannot handle complex shapes with holes, and gets a higher Chamfer loss on Chair. Implicit function method (DVR) produces more accurate reconstructions. While its accuracy is comparable to DEFTET, it takes more than 10 seconds to post-process the prediction. We get similar performance to DVR but win in terms of inference speed.

## 5.3 Tetrahedral Meshing

We compare DEFTET to existing, non-learning based work on tetrahedral meshing, which tries to tetrahedralize the interior of the provided watertight surface mesh. Note that these approaches are not differentiable and cannot be used in other applications investigated in our work.

**Experimental Setting:** We tetrahedralize all chair objects in the test set, 1685 in total. We compare our DEFTET to three state-of-the-art tetrahedral meshing methods: TetGen [44], QuarTet [10] and TetWild [17]. We employ two sets of metrics to compare performance. The first set focuses on the generated tetrahedron quality, by measuring the distortion of each tetrahedron compared to the regular tetrahedron. We follow TetWild [17] and compute six distortion metrics. The other set of metrics focuses on the distance between ground truth surfaces and the generated tetrahedral surfaces.

**Results:** Results are presented in Table 3. We show two representative distortion metrics in the main paper, results with other metrics are provided in the Appendix. We show two qualitative tetrahedral meshing results in Fig. 6. More can be found in the Appendix. Our method achieves comparable mesh quality compared to QuarTet, and outperforms TetGen and TetWild, in terms of distortion metrics. We also achieve comparable performance compared to TetGen and TetWild, and outperform QuarTet, in terms of distance metrics. When utilizing a neural network, we significantly speedup the tetrahedral meshing process while achieving similar performance. Note that TetGen failed on 13 shapes and QuarTet failed on 2 shapes, while TetWild and our DEFTET succeeded on all shapes.

## 5.4 Novel View Synthesis and Multi-view 3D Reconstruction

We showcase DEFTET's efficiency in reconstructing 3D scene geometry from multiple views, and in rendering of novel views.

**Experimental Settings:** We validate our method on the dataset from Nerf [35]. We evaluate DEFTET on the lego, hotdog and chair scenes and use the same data split as in [35]. We evaluate the

| | # Failed /all cases | Distance | | Distortion | | Run Time (sec.) |
|---|---|---|---|---|---|---|
| | | Hausdorff (e-6) ↓ | Chamfer (e-3) ↓ | Min Dihedral ↑ | AMIPS ↓ | |
| Oracle | - | - | - | 70.52 | 3.00 | - |
| TetGen [44] | 13 / 1685 | **2.30** | **4.06** | 40.35 | 4.42 | **38.26** |
| QuarTet [10] | 2 / 1685 | 9810.30 | 11.44 | **54.91** | **3.38** | 77.38 |
| TetWild [17] | 0 / 1685 | 34.66 | 4.08 | 44.73 | 3.78 | 359.29 |
| DEFTET w.o. learning | 0 / 1685 | 453.72 | 4.21 | 54.27 | 3.37 | 1956.80 |
| DEFTET with learning | 0 / 1685 | 440.11 | 4.27 | 50.33 | 3.61 | 49.13 |

Table 3: Quantitative comparisons on **tetrahedral meshing** in terms of different metrics. Closer to the Oracle's distortion (no distortion) indicates better performance.

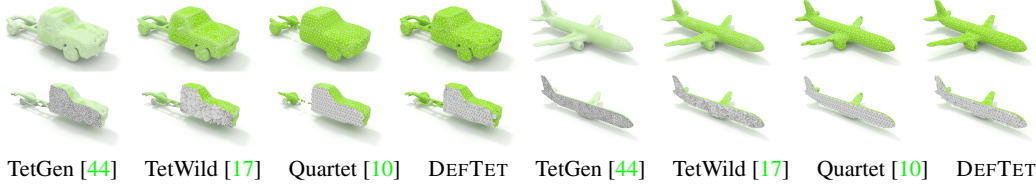

TetGen [44]    TetWild [17]    Quartet [10]    DEFTET    TetGen [44]    TetWild [17]    Quartet [10]    DEFTET

Figure 6: Qualitative results on **tetrahedral meshing**. Please zoom in to see details.

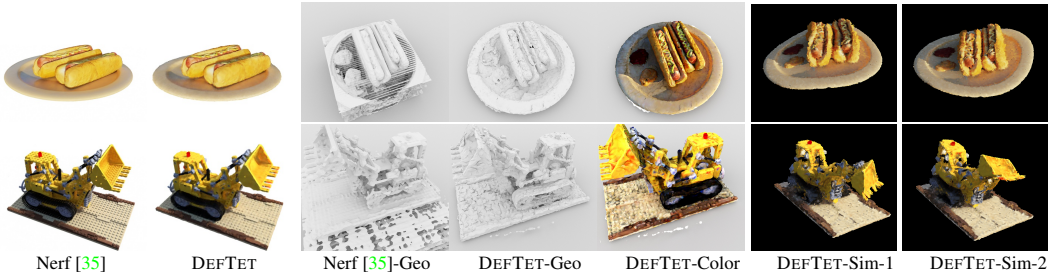

Nerf [35]    DEFTET    Nerf [35]-Geo    DEFTET-Geo    DEFTET-Color    DEFTET-Sim-1    DEFTET-Sim-2

Figure 7: Columns 1-2 show novel view synthesis results on a test view. Columns 3-5 show the reconstructed geometry from Nerf [35], DEFTET, and the reconstructed color of DEFTET. To additionally showcase the power of DEFTET, the last two columns show DEFTET's prediction subjected to physics-based simulation.

quality of rendered views in terms of PSNR, and the exported geometries in terms of the Chamfer Distance, with respect to the training time on NVIDIA RTX 2080 GPU.

**Results:** We report the results of hotdog and lego in Fig. 8 and 7, other results are in Appendix. Our DEFTET converges significantly faster (421 seconds on average) than Nerf [35] (36,000 seconds, around 10 hours), since we do not train a complex neural network but directly optimize vertices colors, occupancies and positions. The drawback is a lower PSNR at converge time as we use vertex colors while Nerf uses neural features. However, to reach the same level of quality, we use significantly less time than Nerf. E.g. to reach 30 PSNR, Nerf needs to take at least half an hour while we need 3 or 4 minutes. Further more, DEFTET generates geometries with materials that can be directly used in other applications, e.g. physics-based simulation.

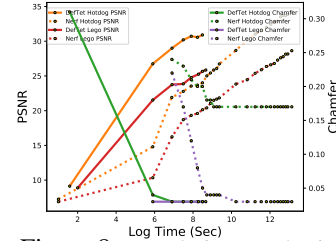

Figure 8: Novel view synthesis: Nerf vs DEFTET.

**Limitations of DEFTET:** We discuss a few limitations to be solved in future work. First, non-flipping of the tetrahedrons is not guaranteed by the prediction of a neural network – we only partially solve this by introducing regularization losses. Second, to represent high fidelity 3D shape, ideas from Oct-trees can be borrowed. Third, our current representation of colors is relatively naive and thus improvements for the novel view synthesis application are possible, bridging the gap in the PSNR metric with the current state-of-the-art approach Nerf [35].

## 6    Conclusion

In this paper, we introduced a parameterization called DEFTET of solid tetrahedral meshes for the 3D reconstruction problem. Our approach optimizes for vertex placement and occupancy and is differentiable with respect to standard 3D reconstruction loss functions. By utilizing neural networks, our approach is the first to produce tetrahedral meshes directly from noisy point clouds and single images. Our work achieves comparable or higher reconstruction quality while running significantly faster than existing work that achieves similar quality. We believe that our DEFTET can be successfully applied to a wide variety of other 3D applications.

## Broader Impact

In this work, we propose and investigate a method for tetrahedral meshing of shapes. The method can be used to mesh arbitrary objects both from point clouds and images. Robust tetrahedral meshing is a longstanding problem in graphics and we showed how a new representation which is amenable to learning can achieve significant advancement in this field. There are several tetrahedral meshing methods that exist in the literature that graphics people or artists use to convert regular meshes into tetrahedral meshes. We hope our work to set the new standard in this domain. Our work targets 3D reconstruction which is a widely studied task in computer vision, robotics and machine learning. Like other work in this domain, our approach can be employed in various applications such as VR/AR, simulation and robotics. We do not anticipate ethics issues with our work.

## Acknowledgments and Disclosure of Funding

We would like to acknowledge contributions through helpful discussions and technical support from Clement Fuji Tsang, Jean-Francois Lafleche, and Joey Litalien. This work was fully funded by NVIDIA and no third-party funding was used.

## Footnotes

[1] For example, all ShapeNet [4] objects can be normalized into unit cube.

[2] https://en.wikipedia.org/wiki/Tetrahedron

[3] https://github.com/crawforddoran/quartet

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
