[Supplementary Material]

# Appendix: Learning Deformable Tetrahedral Meshes for 3D Reconstruction

Jun Gao[1,2,3]    Wenzheng Chen[1,2,3]    Tommy Xiang[1,2]

Alec Jacobson[2]    Morgan McGuire[1]    Sanja Fidler[1,2,3]

NVIDIA[1]    University of Toronto[2]    Vector Institute[3]

{jung, wenzchen, txiang, mcguire, sfidler}@nvidia.com, jacobson@cs.toronto.edu

In Appendix, we first provide details and ablations about the regularization terms we employed in training our DEFTET in Sec. 1. Details about the network architecture we used in different applications are provided in Sec. 2. We show additional experimental details and results in Sec. 3.

## 1 Explanations and Ablations of Loss Functions

In this section, we first provide explanations about different regularizers we employed in the main paper, and we provide ablation studies of different regularization terms. The notation follows the main paper.

### 1.1 Explanations

The Laplacian loss, $L_{\text{lap}}$, penalizes the change in relative positions of neighboring vertices to encourage the deformed tetrahedron to keep the initial regularity:

$$L_{\text{lap}} = \frac{1}{N} \sum_{i=1}^{N} \left| \Delta_{v_i} - \frac{1}{|\mathcal{N}(v_i)|} \sum_{v' \in \mathcal{N}(v_i)} \Delta_{v'} \right|^2, \tag{1}$$

where $\mathcal{N}(v_i)$ denotes the neighbors of vertex $v_i$, and $\Delta_{v_i} = [\Delta x_i, \Delta y_i, \Delta z_i]^T$.

The Delta Loss, $L_{\text{del}}$, encourages the deformation to be local by penalizing large deformation for each of the vertices:

$$L_{\text{del}} = \frac{1}{N} \sum_{i=1}^{N} |\Delta_{v_i}|^2. \tag{2}$$

We want the tetrahedral mesh to remain as regular as possible.

The EquiVolume loss, $L_{\text{vol}}$, encourages each deformed tetrahedrons to have similar volume and avoids flipped tetrahedrons, whose volume is negative, by penalizing the differences in volume for each tetrahedron:

$$L_{\text{vol}} = \frac{1}{K} \sum_{k=1}^{K} |V_k - \overline{V}|^4, \tag{3}$$

where $V_k$ is the volume of tetrahedron $T_k$ and $\overline{V}$ is the average volume. We empirically find the power of 4 works best.

The AMIPS Loss [8], $L_{\text{amips}}$, penalizes the distortion of each tetrahedron to obtain more regular tetrahedrons and avoid needle-like predictions. We follow Tetwild [11] in defining this loss:

$$L_{\text{amips}} = \frac{1}{K} \sum_{k=1}^{K} \frac{\text{tr}(J_k^T J_k)}{\det(J_k)^{\frac{2}{3}}}, \tag{4}$$

|  |  |  |  |  |  |  |
|---|---|---|---|---|---|---|
| GT | $L_{\text{recons}}$ | $+L_{\text{lap}}$ | $+L_{\text{del}}$ | $+L_{\text{vol}}$ | $+ L_{\text{amips}}$ | $+ L_{\text{sm}}$ |

Figure 1: Ablation study on regularization terms. We sequentially add regularizations from left to right.

| $L_{\text{lap}}$ | $L_{\text{del}}$ | $L_{\text{vol}}$ | $L_{\text{amips}}$ | $L_{\text{sm}}$ | Airplane | Bench | Dresser | Car | Chair | Display | Lamp | Speaker | Rifle | Sofa | Table | Phone | Vessel | **Mean** |
|---|---|---|---|---|---|---|---|---|---|---|---|---|---|---|---|---|---|---|
| ✗ | ✗ | ✗ | ✗ | ✗ | 68.93 | 60.43 | 84.50 | 87.02 | 69.61 | 72.15 | 47.28 | 85.45 | 55.91 | 83.92 | 63.75 | 85.00 | 71.97 | 71.99 |
|  | ✗ | ✗ | ✗ | ✗ | 51.01 | 41.03 | 80.52 | 80.23 | 65.39 | 62.75 | 37.97 | 82.18 | 42.61 | 76.02 | 51.62 | 83.92 | 59.52 | 62.67 |
|  |  | ✗ | ✗ | ✗ | 49.20 | 48.14 | 84.14 | 80.10 | 66.41 | 74.18 | 40.34 | 85.22 | 41.53 | 83.02 | 64.57 | 85.88 | 58.75 | 66.27 |
|  |  |  | ✗ | ✗ | 73.10 | 63.64 | 85.53 | 88.45 | 72.59 | 76.38 | 50.92 | 86.70 | 55.73 | 85.58 | 69.56 | 87.13 | 74.89 | 74.63 |
|  |  |  |  | ✗ | 74.52 | 63.53 | 85.78 | 89.06 | 72.79 | 76.42 | 50.97 | 86.60 | 58.44 | 85.96 | 70.36 | 89.28 | 75.17 | 75.30 |

Table 1: Ablation study on **Point Cloud reconstruction**. We report 3D IOU (higher is better).

| $L_{\text{lap}}$ | $L_{\text{del}}$ | $L_{\text{vol}}$ | $L_{\text{amips}}$ | $L_{\text{sm}}$ | Airplane | Bench | Dresser | Car | Chair | Display | Lamp | Speaker | Rifle | Sofa | Table | Phone | Vessel | **Mean** |
|---|---|---|---|---|---|---|---|---|---|---|---|---|---|---|---|---|---|---|
| ✗ | ✗ | ✗ | ✗ | ✗ | 0.68 | 0.63 | 0.61 | 0.57 | 0.82 | 0.82 | 2.57 | 0.80 | 0.76 | 0.63 | 0.83 | 0.46 | 0.87 | 0.85 |
|  | ✗ | ✗ | ✗ | ✗ | 0.62 | 0.58 | 0.54 | 0.56 | 0.71 | 0.71 | 1.70 | 0.73 | 0.61 | 0.56 | 0.68 | 0.40 | 0.75 | 0.70 |
|  |  | ✗ | ✗ | ✗ | 0.65 | 0.61 | 0.54 | 0.55 | 0.70 | 0.71 | 1.80 | 0.72 | 0.68 | 0.56 | 0.70 | 0.40 | 0.77 | 0.72 |
|  |  |  | ✗ | ✗ | 0.60 | 0.58 | 0.58 | 0.52 | 0.72 | 0.71 | 1.82 | 0.74 | 0.74 | 0.58 | 0.71 | 0.42 | 0.75 | 0.73 |
|  |  |  |  | ✗ | 0.56 | 0.58 | 0.57 | 0.51 | 0.71 | 0.72 | 1.82 | 0.74 | 0.75 | 0.56 | 0.70 | 0.38 | 0.73 | 0.72 |

Table 2: Ablation study on **Point Cloud reconstruction**. We report Chamfer Distance (lower is better).

where $J_k$ is the Jacobian of the 3D deformation that transforms the deformed tetrahedron $T_k$ into a regular tetrahedron, $\text{tr}()$ denotes the trace of a matrix, and $\det()$ denotes the matrix determinant.

Smoothness loss [3], $L_{\text{sm}}$, encourages the mesh surface to be smooth by penalizing the difference in normals for neighboring faces. This loss is applied only when 3D ground-truth is available. Let $\mathbf{E}$ be the set of all edges in $\mathcal{F}$, which denotes the set of all triangular faces whose $P_s(v)$ equals to one, and $\theta_i$ be the angle between two neighboring faces, which share the edge $e_i$:

$$L_{\text{sm}} = \frac{1}{|\mathbf{E}|} \sum_{e_i \in \mathbf{E}} (-\cos(\theta_i) + 1)^2. \tag{5}$$

We want the neighbouring boundary faces to be approximately parallel.

We find that all of these regularizations are necessary to obtain non-degenerated and smooth predictions, as shown next.

## 1.2 Ablation Study

We first provide a qualitative ablation study for different terms by optimizing DEFTET to ground truth shape in Fig. 1. We also provide a quantitative evaluation for the task of Point Cloud 3D Reconstruction in Table 1 and 2. We sequentially add regularizations in both experiments. If we only use $L_{\text{recons}}$ loss without any regularizations, the predicted shapes have many flipped and degenerated triangles, and achieve the worst performance in terms of Chamfer Distance. $L_{\text{lap}}$ helps maintain the regularity from the initial tetrahedron. However, although it gives the best performance in terms of Chamfer Distance, the 3D IOU becomes worst. We suspect the reason might because the relatively large deformation of each tetrahedron makes the occupancy prediction harder. $L_{\text{del}}$ encourages local deformation and further helps to maintain the regularity, making the 3D IOU performance better. $L_{\text{vol}}$ helps avoid flipped tetrahedrons, and $L_{\text{amips}}$ helps avoid needle-like predictions. These two regularizations are the main terms to avoid the degenerations in the output and produce results with much better performance in terms of 3D IOU and comparable performance in terms of Chamfer Distance. $L_{\text{sm}}$ further helps to smooth the prediction and make the output look visually better.

## 2 Network Architecture

In this section, we provide details about the network architecture for each of the applications.

**Point Cloud 3D Reconstruction:** For the point cloud encoder, we adopt PVCNN [16], which outputs multi-scale feature maps of size: $C_1 \times 32 \times 32 \times 32, C_2 \times 16 \times 16 \times 16, C_3 \times 8 \times 8 \times 8,$

respectively, where $C_1, C_2, C_3$ are the number of channels and set to $C_1 = 64, C_2 = 256, C_3 = 512$. We have two encoders: one encoder provides a feature map for the Pos-Decoder, which predicts the deformation for each vertex. The second encoder provides a separate feature map for the Occ-Decoder, which predicts the occupancy for each tetrahedron. The Pos-Decoder is composed of 2 Graph-Convolutional-Layers [15] (GCN), followed by 2 Multi-Layer-Perceptron (MLP) layers. Each GCN layer has 256/128 hidden neurons, respectively, and each MLP layer has 128/64 hidden neurons, respectively. The Occ-Decoder comprises of four MLP layers, with hidden neurons being 256, 256, 128, and 64, respectively.

**Single Image 3D Reconstruction using 3D Supervision:** We adopt the network architecture from DISN [25]. In particular, we have two VGG16 [22] image encoders, one is for the prediction of occupancy of each tetrahedron, and the other is for the prediction of deformation of each vertex. For each encoder, we have a corresponding decoder to predict the occupancy and deformation, respectively. For the deformation prediction decoder, we simply change the output dimension from one to three. We refer to the original DISN [25] paper for more details about the architecture.

**Single Image 3D Reconstruction using 2D Supervision:** For a fair comparison, we adopt the network architecture from DVR [20] but change the output of the last layer to accommodate the tetrahedron shape. We use $40 \times 40 \times 40$ resolution in 2D supervision training as we find it is enough to provide a high-quality reconstruction with relatively low memory cost. Note that higher resolution could further improve the results.

**Laplacian Layer:** The Laplacian Layer is a differentiable layer, where the position of each vertex computes the average of its neighboring boundary vertices. Specifically, for the boundary vertex $v_i$ in the prediction:

$$v_i = \frac{1}{|\mathcal{N}'|} \sum_{v' \in \mathcal{N}'(v_i)} v', \tag{6}$$

where $\mathcal{N}'$ denotes all the vertices that are both on the predicted boundary surface and neighbouring to $v_i$. To train the model with a Laplacian layer, we first pre-train the occupancy and deformation prediction, and have an additional Def-Decoder to predict a new position offset for each vertex. The Laplacian layer is applied after this prediction, and the loss function is applied to the smoothed vertices. Since the layer is differentiable, we can back-propagate the gradient to the network to learn to adapt to the Laplacian Smoothing. Another approach to incorporate laplacian smoothing is through the loss function, where

$$L_{\text{lap-sm}} = \frac{1}{N} \sum_{i=1}^{N} w_i \left| v_i - \frac{1}{|\mathcal{N}'|} \sum_{v' \in \mathcal{N}'(v_i)} v' \right|, \tag{7}$$

We filter all the vertices that are far away from the predicted boundary surface by setting $w_i = 0$, and $w_i = 1$ if the vertex is on the boundary. We then penalize the difference between its position to the average of the positions of its neighbour boundary vertices. We encourage the position of $v_i$ to lie in the center of its neighbouring boundary vertices.

**Novel view synthesis and Multi-view 3D Reconstruction:** In this application, we directly optimize the position offset, occupancy, and RGB color for each vertex. Introducing neural networks to further regularize the vertex color, position or occupancy would be an interesting direction, but we leave it for future work.

## 3 Experimental Details and Additional Results

We first describe the pipeline to prepare the ShapeNet [2] dataset for training in Sec. 3.1, and report experimental details and additional results for each application in the following sections.

### 3.1 Data Preparation

Since many ShapeNet objects [2] are not watertight, we follow the pipeline in Kaolin [13] to convert all the objects to a watertight surface mesh. Note that other methods [12] for this procedure can be

| Category | Airplane | Bench | Dresser | Car | Chair | Display | Lamp | Speaker | Rifle | Sofa | Table | Phone | Vessel | Mean↓ | Time(ms)↓ |
|---|---|---|---|---|---|---|---|---|---|---|---|---|---|---|---|
| 3D-R2N2 [4] | 0.89 | 0.96 | 0.92 | 0.89 | 1.02 | 1.13 | 2.99 | 1.02 | 0.85 | 0.93 | 1.14 | 0.93 | 1.17 | 1.14 | 174.49 |
| DeepMCube [14] | 1.60 | 1.48 | 1.21 | 1.33 | 1.68 | 1.77 | 2.74 | 1.71 | 1.31 | 1.53 | 1.65 | 0.89 | 1.87 | 1.60 | 349.83 |
| Pixel2mesh [24] | **0.48** | 0.67 | 0.66 | 0.55 | 0.93 | 0.72 | **1.29** | 0.79 | **0.45** | 0.58 | 1.05 | 0.37 | **0.62** | **0.71** | 30.31 |
| OccNet [18] | 1.15 | 1.34 | 1.09 | 1.16 | 2.10 | 1.25 | 3.34 | 1.54 | 1.17 | 1.19 | 1.55 | 0.76 | 1.90 | 1.51 | 728.36 |
| OccNet - Our Arch | 0.51 | 0.56 | **0.50** | 0.46 | 0.77 | 0.84 | 3.06 | 0.76 | 0.67 | 0.53 | 0.85 | 0.37 | 0.77 | 0.83 | 866.54 |
| MeshRCNN [9] | 0.56 | 0.61 | 0.62 | 0.53 | 0.76 | 0.89 | 2.01 | 1.17 | 0.61 | 0.68 | 0.96 | 0.43 | 0.82 | 0.82 | 228.46 |
| FIXEDTET | 0.70 | 0.72 | 0.72 | 0.66 | 1.03 | 1.01 | 3.64 | 0.99 | 0.91 | 0.76 | 1.07 | 0.56 | 1.05 | 1.07 | 43.52 |
| DEFTET | 0.55 | **0.54** | 0.52 | **0.48** | **0.69** | **0.66** | 2.08 | **0.75** | 0.80 | **0.49** | **0.65** | **0.32** | 0.78 | **0.72** | 61.39 |

Table 3: Results on **Point Cloud Reconstruction**. We report Chamfer Distance (e-2), lower is better.

| Category | Airplane | Bench | Dresser | Car | Chair | Display | Lamp | Speaker | Rifle | Sofa | Table | Phone | Vessel | Mean↓ | Time(ms)↓ |
|---|---|---|---|---|---|---|---|---|---|---|---|---|---|---|---|
| 3D-R2N2 [4] | 2.34 | 2.45 | 2.37 | 2.34 | 2.68 | 2.84 | 7.90 | 2.67 | 2.15 | 2.40 | 2.91 | 2.29 | 2.97 | 2.95 | 174.49 |
| DeepMCube [14] | 4.30 | 3.80 | 3.03 | 3.52 | 4.40 | 4.44 | 7.57 | 4.38 | 3.42 | 3.91 | 4.13 | 2.28 | 4.93 | 4.16 | 349.83 |
| Pixel2mesh [24] | **1.35** | 1.84 | 1.80 | 1.55 | 2.53 | 1.97 | **3.65** | 2.21 | **1.26** | 1.62 | 2.76 | 1.06 | **1.75** | 1.95 | **30.31** |
| OccNet [18] | 3.13 | 3.51 | 2.75 | 3.06 | 5.54 | 3.25 | 9.17 | 3.98 | 3.09 | 3.09 | 3.95 | 2.01 | 4.98 | 3.9 | 728.36 |
| OccNet - Our Arch | 1.44 | 1.57 | **1.43** | **1.33** | 2.16 | 2.28 | 8.48 | 2.12 | 1.79 | 1.48 | 2.28 | 1.06 | 2.13 | 2.27 | 866.54 |
| MeshRCNN [9] | 1.54 | 1.65 | 1.69 | 1.47 | 2.07 | 2.32 | 5.44 | 3.02 | 1.62 | 1.82 | 2.47 | 1.19 | 2.19 | 2.19 | 228.46 |
| FIXEDTET | 1.96 | 2.00 | 2.00 | 1.88 | 2.86 | 2.62 | 7.01 | 2.73 | 2.44 | 2.11 | 2.87 | 1.54 | 2.90 | 2.69 | 43.52 |
| DEFTET | 1.56 | **1.50** | 1.46 | 1.37 | **1.95** | **1.82** | 5.81 | **2.10** | 2.15 | **1.41** | **1.81** | **0.92** | 2.16 | **2.00** | 61.39 |

Table 4: Results on **Point Cloud Reconstruction**. We report Chamfer L1 (e-2), lower is better.

also employed, and we choose Kaolin for its simplicity. In particular, each object is first voxelized to a high-resolution voxel representation (in our case, $100 \times 100 \times 100$), and the surface mesh is then obtained via the Marching Cube algotrithm [17], followed by Laplacian Smoothing [10].

## 3.2 Point Cloud 3D Reconstruction

**Experimental Details:** To prepare the input for point cloud 3D reconstruction, we randomly sample 5000 points from the object's surface and add Gaussian noise, whose variance is set to 0.5% of the unit cube length. For each category, we randomly select 70% shapes to train the network, 5% for validation, and remaining 25% for testing. The hyper-parameters we used in the loss function are: $\lambda_{recon} = 1, \lambda_{surf} = 10, \lambda_{vol} = 1, \lambda_{lap} = 1e-3, \lambda_{sm} = 1e-2, \lambda_{del} = 1e-3, \lambda_{amips} = 1e-5$.

To evaluate different methods, we have several metrics: 3D Intersection-Over-Union (IOU), Chamfer Distance and Chamfer-L1. For 3D IOU, we follow OccNet [18] and randomly sample 100k points in 3D space to evaluate ground truth occupancies and check whether the points are inside or outside of the predicted mesh. The definition of Chamfer and Chamfer-L1 Distance is:

$$\text{Chamfer}(\mathcal{F}_{\text{Pred}}, \mathcal{F}_{\text{GT}}) = \frac{1}{2|\partial\mathcal{F}_{\text{Pred}}|} \sum_{p\in\partial\mathcal{F}_{\text{Pred}}} \min_{q\in\partial\mathcal{F}_{\text{GT}}} ||p-q||_2 + \frac{1}{2|\partial\mathcal{F}_{\text{GT}}|} \sum_{p\in\partial\mathcal{F}_{\text{GT}}} \min_{q\in\partial\mathcal{F}_{\text{Pred}}} ||p-q||_2, \quad (8)$$

$$\text{Chamfer-L1}(\mathcal{F}_{\text{Pred}}, \mathcal{F}_{\text{GT}}) = \frac{1}{|\partial\mathcal{F}_{\text{Pred}}|} \sum_{p\in\partial\mathcal{F}_{\text{Pred}}} \min_{q\in\partial\mathcal{F}_{\text{GT}}} ||p-q||_1 + \frac{1}{|\partial\mathcal{F}_{\text{GT}}|} \sum_{p\in\partial\mathcal{F}_{\text{GT}}} \min_{q\in\partial\mathcal{F}_{\text{Pred}}} ||p-q||_1, \quad (9)$$

where $\mathcal{F}_{\text{GT}}, \mathcal{F}_{\text{Pred}}$ denote all the faces in the ground truth and prediction, respectively, and $\partial\mathcal{F}_{\text{GT}}, \partial\mathcal{F}_{\text{Pred}}$ denote the set of sampled points in the ground truth, and prediction, respectively. Since Chamfer Distance and Chamfer-L1 lead to noisy estimates when sampling on the surface, we further introduce a more accurate distance metric: Hausdorff-Avg Distance, an extension of the Hausdorff Distance [5, 1, 23] from Computer Graphics to compute the shape similarity:

$$\text{Hausdorff-Avg}(\mathcal{F}_{\text{Pred}}, \mathcal{F}_{\text{GT}}) = \frac{1}{2|\partial\mathcal{F}_{\text{Pred}}|} \sum_{p\in\partial\mathcal{F}_{\text{Pred}}} \min_{f\in\mathcal{F}_{\text{GT}}} \text{dist}_f(p, f) + \frac{1}{2|\partial\mathcal{F}_{\text{GT}}|} \sum_{p\in\partial\mathcal{F}_{\text{GT}}} \min_{f\in\mathcal{F}_{\text{Pred}}} \text{dist}_f(p, f), \quad (10)$$

where $\text{dist}_f(p, f)$ is the point-surface distance from point $p$ to face $f$.

**Additional Experimental Results:** Table 3, 4, and 5 show the quantitative results in terms of Chamfer Distance, Chamfer-L1, and Hausdorff-Avg, respectively. We also provide additional qualitative examples in Fig. 2 and 3. We achieve better performance compared with voxel-based and occupancy-based representations, and Deep Marching Cubes [14]. Compared to a mesh-based representation, we perform significantly better in the shape categories that have holes, such as Chair and Table, while the average accuracies over all the categories are similar.

| Category | Airplane | Bench | Dresser | Car | Chair | Display | Lamp | Speaker | Rifle | Sofa | Table | Phone | Vessel | Mean↓ | Time(ms)↓ |
|---|---|---|---|---|---|---|---|---|---|---|---|---|---|---|---|
| 3D-R2N2 [4] | 0.86 | 0.92 | 0.86 | 0.85 | 0.96 | 1.08 | 2.96 | 0.95 | 0.83 | 0.87 | 1.09 | 0.89 | 1.14 | 1.09 | 174.49 |
| DeepMCube [14] | 1.58 | 1.45 | 1.15 | 1.29 | 1.64 | 1.73 | 2.72 | 1.64 | 1.30 | 1.49 | 1.61 | 0.85 | 1.85 | 1.56 | 349.83 |
| Pixel2mesh [24] | **0.44** | 0.62 | 0.58 | 0.49 | 0.86 | 0.65 | **1.25** | 0.71 | **0.43** | 0.51 | 0.99 | 0.31 | 0.59 | **0.65** | 30.31 |
| OccNet [18] | 1.13 | 1.30 | 1.02 | 1.13 | 2.06 | 1.20 | 3.32 | 1.47 | 1.16 | 1.14 | 1.49 | 0.73 | 1.89 | 1.46 | 728.36 |
| OccNet - Our Arch | 0.47 | 0.51 | **0.41** | **0.40** | 0.70 | 0.77 | 3.01 | 0.68 | 0.66 | 0.45 | 0.78 | 0.30 | **0.73** | 0.76 | 866.54 |
| MeshRCNN [9] | 0.53 | 0.56 | 0.53 | 0.46 | 0.70 | 0.83 | 1.97 | 1.08 | 0.59 | 0.61 | 0.89 | 0.37 | 0.79 | 0.76 | 228.46 |
| FIXEDTET | 0.66 | 0.66 | 0.63 | 0.60 | 0.95 | 0.89 | 2.45 | 0.90 | 0.89 | 0.68 | 0.99 | 0.50 | 1.02 | 0.91 | 43.52 |
| DEFTET | 0.52 | **0.48** | **0.41** | **0.41** | **0.61** | **0.58** | 2.04 | **0.65** | 0.78 | **0.41** | **0.56** | **0.24** | 0.74 | **0.65** | 61.39 |

Table 5: Results on **Point Cloud Reconstruction**. We report Hausdorff-Avg (e-2), lower is better.

## 3.3 Single Image 3D Reconstruction using 3D Supervision

**Experimental Settings:** We use the same dataset as in DVR [20], and use the same training, validation, and testing split. For the ground truth 3D shape, we use the same as described in Sec. 3.1. We train one network on all 13 ShapeNet core categories. We compare DEFTET with voxel-based representation [4], mesh-based representation [24], and implicit function representations [25]. We additionally compare with Deep Marching Cubes [14]. For 3DR2N2 [4], Pixel2Mesh [24] and Deep Marching Cube [14], we use the official codebase[1]. We reimplemented DISN [25] in PyTorch. The hyperparameters for DEFTET follow those in Point Cloud 3D Reconstruction. We evaluate all these methods using 3D IOU, Chamfer, Chamfer-L1 and Hausdorff-Avg Distance.

**Experimental Results:** Table 6, 7, 8, and 9 show quantitative results in terms of 3D IOU, Chamfer Distance, Chamfer-L1, and Hausdorff-Avg, respectively. We also provide qualitative examples in Fig. 4 and 5. We achieve similar performance compared to the implicit-function-based representation in terms of 3D IOU, and significantly outperform other representations in terms of other metrics, since we explicitly deform the vertices to better align with the surface. Compared to a mesh-based representation, we perform significantly better in the shape categories that have holes, such as Chair and Table.

| Category | Airplane | Bench | Dresser | Car | Chair | Display | Lamp | Speaker | Rifle | Sofa | Table | Phone | Vessel | Mean | Time(ms)↓ |
|---|---|---|---|---|---|---|---|---|---|---|---|---|---|---|---|
| 3D-R2N2 [4] | 39.99 | 32.68 | 66.13 | 67.82 | 39.41 | 39.90 | 32.92 | 63.23 | 28.37 | 57.93 | 43.56 | 61.47 | 41.37 | 47.29 | **13.41** |
| DeepMCube [14] | 19.06 | 10.17 | 44.79 | 40.91 | 19.28 | 26.15 | 19.73 | 47.77 | 19.61 | 30.81 | 18.38 | 26.10 | 26.30 | 26.85 | 329.29 |
| Pixel2mesh [24] | 49.84 | 31.89 | 66.18 | 73.53 | 36.61 | 42.68 | 31.64 | 65.67 | 27.37 | 62.88 | 38.34 | 61.91 | 45.48 | 48.77 | 25.85 |
| DISN [25] | **56.31** | 36.12 | 71.72 | 76.67 | 47.57 | 47.62 | **40.55** | 71.82 | **34.30** | 68.39 | 52.07 | 64.93 | **50.82** | **55.30** | 1363.58 |
| DEFTET | 55.44 | **36.40** | **73.46** | **77.38** | 47.51 | 47.48 | 37.94 | **73.48** | 31.20 | **69.33** | **53.11** | **66.89** | 48.44 | 55.24 | 200.63 |

Table 6: Results on **Single Image 3D Reconstruction** with **3D supervision**. We report 3D IOU.

| Category | Airplane | Bench | Dresser | Car | Chair | Display | Lamp | Speaker | Rifle | Sofa | Table | Phone | Vessel | Mean | Time(ms)↓ |
|---|---|---|---|---|---|---|---|---|---|---|---|---|---|---|---|
| 3D-R2N2 [4] | 2.26 | 2.00 | 2.02 | 1.80 | 2.83 | 3.01 | 4.33 | 2.94 | 2.26 | 2.38 | 2.17 | 1.78 | 2.69 | 2.50 | 13.41 |
| DeepMCube [14] | 4.80 | 7.58 | 5.50 | 5.79 | 7.01 | 6.78 | 6.39 | 6.73 | 3.62 | 7.40 | 6.10 | 6.48 | 5.23 | 6.11 | 329.29 |
| Pixel2mesh [24] | 1.52 | **1.62** | 1.85 | 1.30 | 2.64 | 2.56 | 2.91 | 2.67 | **1.82** | 1.90 | 2.20 | 1.59 | **2.01** | 2.04 | 25.85 |
| DISN [25] | 1.52 | 1.96 | 1.61 | 1.28 | 2.51 | **2.48** | **3.49** | 2.21 | 2.15 | 1.66 | 1.78 | 1.55 | 2.29 | 2.04 | 1363.58 |
| DEFTET | **1.49** | 1.77 | **1.44** | **1.18** | **2.39** | 2.52 | 3.53 | **2.03** | 2.13 | **1.58** | **1.68** | **1.34** | 2.26 | **1.95** | 200.63 |

Table 7: Results on **Single Image 3D Reconstruction** with **3D supervision**. We report Chamfer Distance.

| Category | Airplane | Bench | Dresser | Car | Chair | Display | Lamp | Speaker | Rifle | Sofa | Table | Phone | Vessel | Mean | Time(ms)↓ |
|---|---|---|---|---|---|---|---|---|---|---|---|---|---|---|---|
| 3D-R2N2 [4] | 6.01 | 5.03 | 4.85 | 4.66 | 7.30 | 7.47 | 11.61 | 7.37 | 5.68 | 5.93 | 5.41 | 4.44 | 6.94 | 6.366 | 13.41 |
| DeepMCube [14] | 13.15 | 18.92 | 12.98 | 14.79 | 17.96 | 16.44 | 17.40 | 16.40 | 9.33 | 18.25 | 15.07 | 15.17 | 13.78 | 15.36 | 329.29 |
| Pixel2mesh [24] | 4.23 | **4.24** | 4.60 | 3.52 | 6.95 | 6.49 | 8.15 | 6.85 | **4.83** | 4.85 | 5.61 | 4.01 | **5.49** | 5.37 | 25.85 |
| DISN [25] | 4.19 | 5.20 | 4.06 | 3.38 | 6.66 | **6.32** | 9.63 | 5.66 | 5.63 | 4.29 | 4.55 | 3.90 | 6.07 | 5.35 | 1363.58 |
| DEFTET | **4.11** | 4.70 | **3.69** | **3.16** | **6.38** | 6.40 | 9.85 | **5.31** | 5.44 | **4.09** | **4.35** | **3.42** | 6.06 | **5.15** | 200.63 |

Table 8: Results on **Single Image 3D Reconstruction** with **3D supervision**. We report Chamfer L1.

| Category | Airplane | Bench | Dresser | Car | Chair | Display | Lamp | Speaker | Rifle | Sofa | Table | Phone | Vessel | Mean | Time(ms)↓ |
|---|---|---|---|---|---|---|---|---|---|---|---|---|---|---|---|
| 3D-R2N2 [4] | 2.24 | 1.97 | 1.97 | 1.77 | 2.79 | 2.97 | 4.29 | 2.88 | 2.25 | 2.34 | 2.12 | 1.75 | 2.67 | 2.46 | 13.41 |
| DeepMCube [14] | 4.78 | 7.56 | 5.47 | 5.77 | 6.99 | 6.76 | 6.37 | 6.70 | 3.62 | 7.37 | 6.08 | 6.47 | 5.22 | 6.09 | 329.29 |
| Pixel2mesh [24] | 1.49 | **1.59** | 1.80 | 1.26 | 2.60 | 2.52 | 2.87 | 2.62 | **1.80** | 1.86 | 2.16 | 1.56 | **1.98** | 2.01 | 25.85 |
| DISN [25] | 1.50 | 1.93 | 1.56 | 1.25 | 2.48 | **2.45** | **3.46** | 2.15 | 2.14 | 1.62 | 1.73 | 1.52 | 2.28 | 2.01 | 1363.58 |
| DEFTET | **1.46** | 1.74 | **1.38** | **1.14** | **2.35** | 2.48 | 3.51 | **1.96** | 2.12 | **1.54** | **1.63** | **1.31** | 2.24 | **1.91** | 200.63 |

Table 9: Results on **Single Image 3D Reconstruction** with **3D supervision**. We report Mean Hausdorff.

Figure 2: Qualitative results on **Point Cloud 3D Reconstruction**. * denotes original OccNet [18] architecture.

### 3.4 Single Image 3D Reconstruction using 2D Supervision

**Experimental Settings:** We set the hyper-parameters as follows: $\lambda_{\text{recon}} = 2, \lambda_{\text{lap}} = 1e - 3, \lambda_{\text{del}} = 1e - 3, \lambda_{\text{sm}_2 D} = 1, \lambda_{\text{vol}} = 0, \lambda_{\text{amips}} = 0$. In 2D case, we empirically find that $L_{\text{vol}}$ and $L_{\text{amips}}$ does not influence the performance due to our multiple-stage training policy. During training, we first fix the grid, and only predict the occupancy and color. After it starts converging (usually 24 hours), we also predict deformation, occupancy and color for each vertex. The second fine-tuning takes another 24 hours. Finally, we add $L_{\text{lap-sm}}$ to smooth the surface, which takes 10 hours. The total training costs around 60 hours on a RTX 2080 GPU.

**Marching-Tet Layer:** In this task, due to the lack of 3D supervision, we find that applying Marching Tet [6] to extract ISO surface will improve the performance. While Marching Tet has been developed for more than 20 years, we reproduce it in PyTorch and make it differentiable with respect to vertex position and color. Specifically, we adaptively subdivide a tetrahedron based on its occupancy with a given threshold. If the tetrahedron lies across the surface, we cut the edge based on the given threshold to make sure the subdivided tetrahedrons perfectly fit the surface. If the tetrahedron lies inside or outside, we cut the edge in the middle point. Compared to traditional Matching-Tet functions, our implemented Marching-Tet layer takes a tetrahedral mesh as input and returns a subdivided tetrahedral mesh and is fully differentiable, while the traditional Marching Tet [6] returns a triangular mesh and is not differentiable. Moreover, we could also interpolate other vertex attributes like vertex colors.

**Additional Experimental Results:** We show additional qualitative results in Fig. 6 and 7. Compared to DIBR [3], a mesh-based representation, we reconstruct shapes with arbitrary topology. Compared to DVR, an implicit-function-based representation, our shapes are more aligned with the input image. Applying Marching-Tet as a post-processing step shrinks the shape, but the overall structure is similar.

### 3.5 Tetrahedral Meshing

**Additional Distortion Metrics:** We show all distortion metrics in Fig. 8. We achieve comparable performance compared to Quartet [7], and outperform TetWild [11] and TetGen [21] in terms of distortion metrics.

**Additional Examples:** We show additional qualitative results in Fig. 9. Our method reaches comparable tetrahedral meshing quality compared to TetWild [11] and TetGen [21], and is significantly better than QuarTet [7].

### 3.6 Novel View Synthesis and Multi-view 3D Reconstruction

**Experimental Settings:** Since the scenes in the Nerf [19]'s dataset have more significant geometry and texture variations than those in ShapNet [2], we apply subdivision and use higher resolution tetrahedral meshes to represent the complex objects in the Nerf dataset. To be specific, we start with a $40 \times 40 \times 40$ resolution and subdivide the tetrahedral mesh twice, resulting in a $160 \times 160 \times 160$ resolution in the final stage of optimization. Since there is no need to subdivide the transparent tetrahedrons, we first delete tetrahedrons whose occupancy is less than a threshold (0.001 in our experiments) before applying subdivision. We find that mask loss is optional in multi-view 3D reconstruction and ablate its use.

**Additional Experimental Results:** We provide additional results in Fig. 10 and Fig. 11. Compared to Nerf [19], we get smooth novel view rendering effects, due to the use of simple vertex colors and not considering light variations. However, our explicit representation allows us to directly export meshes with materials, which can be further applied to other downstream tasks and edited by the users. We also remind the reader that our method is significantly faster than Nerf.

We ablate the difference of adding mask loss in Fig. 10. While their synthesized images are similar, masks help in predicting much clean geometries.

Figure 3: Qualitative results on **Point Cloud 3D Reconstruction**. The first column shows input point cloud.

| Input Img | 3DR2N2 [4] | Pix2Mesh [24] | DMC [14] | OccNet [18] | DEFTET |

Figure 4: Qualitative results on **Single Image 3D Reconstruction** using **3D supervision**.

Figure 5: Qualitative results on **Single Image 3D Reconstruction** using **3D supervision**. The first column shows input image.

INPUT　　　　DVR [20]　　　　DIBR [3]　　　FixedTet　　　DefTet　　　DefTet+MTet

Figure 6: Qualitative results on **Single Image Reconstruction** with **2D supervision**. Please zoom in to see details.

Figure 7: Qualitative results on **Single Image 2D Reconstruction** using **3D supervision**. First column shows input image.

Figure 8: Comparison on **tetrahedral meshing** in terms of distortion metrics. We show the oracle performance in the last row. Results that are closer to the last row indicate less distortion and better performance.

TetGen [21]    TetWild [11]    Quartet [7]    DEFTET        TetGen [21]    TetWild [11]    Quartet [7]    DEFTET

Figure 9: Qualitative results on **tetrahedral meshing**. Please zoom in to see details.

Figure 10: The columns 1-2 show the novel view synthesis results on test view. The columns 3-5 show the reconstructed geometry from Nerf [19], DEFTET, and the reconstructed color of DEFTET. The columns 6-7 show the reconstructed geometry from DEFTET, and the reconstructed color of DEFTET when having access to ground truth mask.

Figure 11: Novel view synthesis and Multi-view 3D Reconstruction: We compare DEFTET with Nerf [19] for the Chair scene in terms of PSNR and Chamfer Distance. DEFTET converges much faster than Nerf.

## Footnotes

[1] https://github.com/autonomousvision/occupancy_networks