[Reviews · NeurIPS 2020]

Review 1

Summary and Contributions: This work proposes a new approach to parameterise 3D shapes within deep networks, based on deforming volumetric tetrahedral meshes. At its core is the idea of, given a pre-defined tetrahedral mesh, optimising simultaneously for vertex displacement and tetrahedral occupancy.

Strengths: - The proposed parameterisation yields an end-to-end differentiable surface mesh parameterisation, unlike deep implicit fields - Compared to volumetric approaches based on voxel grids, it allows for the generation of higher quality results, given comparable volumetric resolution, as it allows for additional freedom by displacing mesh vertices - Existing differentiable rendering techniques can be easily adapted to the framework, as shown by the authors

Weaknesses: - the training loss function of equation (5) contains several regularisation terms, but their impact of final 3D reconstruction quality is not evaluated qualitatively nor quantitatively. This makes the contribution of each single loss component impossible to quantify. - unlike deep implict fields, this representation is limited in resolution. No study of the impact of resolution on final reconstruction accuracy is provided by the authors. This makes it hard to compare the method to representation that are not limited in resolution, especially given that in several qualitative results (see supplementary section) OccNet seems to deliver higher quality reconstructions. - the presented applications do not highlight the true advantage of the proposed representation, i.e. its end-to-end differentiability, as several differentiable rendering techniques have been developed for deep implicit fields. Studying applications where end-to-end differentiability is necessary would have made the submission stronger, and would have better highlighted its advantages over deep implicit fields.

Correctness: The methodology is correct, and results in an end-to-end differentiable 3d surface parameterization.

Clarity: The paper is well written and easy to follow, although I would have preferred to have some details about the "deforming" neural network architecture (e.g. it uses graph convolutions + mlps) in the main paper.

Relation to Prior Work: Yes, the related work section clarifies how the method differs from concurrent approaches.

Reproducibility: Yes

Additional Feedback: I really enjoyed this work, and looking forward to the authors rebuttal and other reviews. My not so positive overall score is due to the lack of solid ablation studies + a convincing application highlighting the advantage of having an end-to-end differentiable mesh representation. --------------- The rebuttal addresses my concerns thus I upgrade my recommendation to 7


Review 2

Summary and Contributions: The paper proposed a novel data structure named "Deformable Tetrahedral Mesh (DEFTET)" for 3D representation utilizing Tetrahedral meshes. DEFTET is a cube fully subdivided into tetrahedrons, where the 3D shape is volumetric and embedded within the tetrahedral mesh. It shows effectiveness in representing local geometric details for 3D object reconstruction from noisy point clouds and images.

Strengths: 1. The data structure DEFTET for 3D representation is novel, and is suitable for end-to-end deep learning. It's definitely a crucial problem to exploit better 3D representation methods for surface understanding. 2. Experimental results demonstrate that DEFTET can adequately represent the geometric details for 3D reconstruction tasks. 3. The paper is well written and organized.

Weaknesses: The reviewer has some major concerns about the experiments. 1. The paper combines many objectives (about nine loss terms in Eq. 5, Eq. 8, and Eq. 12) to optimize the reconstruction network, but has not studied these losses in the experiments section. Such a complex loss function may weaken the contribution of the data representation. Besides, it seems unfair for the compared methods. Do some of these losses can be used for other methods such as Pixel2mesh and MeshRCNN? 2. The SDF (recent SOTA 3D representation method) based approaches (e.g., DISN [1]) have not been discussed and compared in the submission. 3. While the proposed method can not perform better than existing methods such as Pixel2mesh, MeshRCNN, and DISN [1] for 3D reconstruction from images, the paper has not analyzed the reasons. The reviewer suggests presenting some qualitative results of these SOTA methods in Figure 5. 4. The reviewer suggests showing the smoothed GT shapes in Figure. 3 and Figure. 5 so that the readers can better understand the quality of the reconstruction. A minor concern: 1. For Eq. 9~11, How about directly using the last visible surface? Dose Eq. 10 really improve the performance? For example, if f_1 is partial occluded (D_a is visible), f_2 is visible. The color attribute of the pixel I should mainly depend on f_2, right? Especially in the case that f_1 and f_2 are from different parts (e.g, chair leg and chair body), then why do you directly use the color attributes of f_2. [1] Xu, Qiangeng, Weiyue Wang, Duygu Ceylan, Radomir Mech, and Ulrich Neumann. "Disn: Deep implicit surface network for high-quality single-view 3d reconstruction." In Advances in Neural Information Processing Systems, pp. 492-502. 2019.

Correctness: Yes.

Clarity: Yes.

Relation to Prior Work: Yes.

Reproducibility: No

Additional Feedback: The authors addressed most of my concerns. I am willing to increase my rating to 6. However, as a general audience, I believe we care about both the idea and its effectiveness. From this point, the ablation studies are definitely unsatisfied. For example, without a careful study of the messy loss terms, the audiences may not know how much the proposed DefTet contributes to the reconstruction results. I encourage the authors to comprehensively discuss their method in the experimental section for better presentation.


Review 3

Summary and Contributions: The paper proposes a differentiable representation for 3D shape in the form of a tetrahedral mesh. The main idea is that the unit cube can be subdivided into a preset number of tetrahedra and the latter can adapt to represent the input via operations that displace the vertices and estimate the occupancy probability of each tetrahedron. Several applications and comparisons with recent algorithms are shown demonstrating the flexibility and effectiveness of the representations, which has been dubbed DefTet.

Strengths: S1. The authors argue that a representation of 3D shape in the form of tetrahedra has certain advantages over alternatives due to its ability to handle arbitrary topology, allocate more vertices to regions with surface details, and, unlike implicit functions, represent shapes with sharp features. These advantages are explained clearly and documented well. S2. The formulation that estimates displacements for the vertices and occupancy probabilities for the tetrahedra is novel and potentially very powerful. It is the main argument for accepting the paper. S3. The two reconstruction losses for 3D or 2D supervision are also among the strengths of the paper due to the flexibility they provide. S4. Experimental validation is extensive. Results on par or better than the state of the art are shown on 3D reconstruction from noisy point clouds or singe images, as well as on tetrahedral meshing. Comparisons to strong, recent baselines are included in all cases. The fact that DefTet is competitive in all settings, even it does not always dominate, is a strength. (I would have liked to see results on categories other than chairs, but the experiments have breadth in terms of the number of problems addressed.)

Weaknesses: W1. Without being aware of a better alternative, I find the fact that the number of vertices has to be prespecified and remain fixed somewhat unsatisfactory. Ideally, a representation would adapt to the complexity of the input. A large object would give rise to a point cloud with a large number of points, a detailed surface would require more triangled to capture it with high fidelity, etc. W2. More importantly, it is not clear that this approach would scale to large, complex scenes. All results are on single objects. It seems likely that optimization is not guaranteed to converge to a good solution if the complexity of the input was higher. Training a model to reconstruct multiple objects would also probably be hard. I think that this limitation should be acknowledged in the paper. It should not be used as grounds for rejection, since no other approach matches the advantages of DefTet and is also scalable. W3. The loss function is fairly complex and depends on several hyper-parameters to balance its components. A discussion on how these values can be set and on the sensitivity of the results to them should be provided. The only mention of specific values for the lambdas is in lines 69-70 of the supplement, but no intuition is given on how these values are chosen. W4. The smoothness loss is not motivated and described well. It should be made clearer that it is only applied on surface triangles, however according to the supplement it is applied on all faces in F, which is defined as the set of all faces of the tetrahedralization on p. 3 of the main paper. Each edge of the tetrahedral mesh is shared by multiple triangles, more than two in general, and these cannot be approximately parallel to each other. Each edge on the surface, however, is shared by two triangles which should be approximately parallel in a smooth surface. Equation (5) of the supplement seems wrong since it attains maximum value when theta_i is 0. W5. It seems hard to ensure that no vertex will pierce the triangle formed by the other three vertices of the tetrahedron. A few terms of the loss function penalize deformations that may lead to “flipped tetrahedra” but no guarantee is provided. I understand that providing such a guarantee is hard, but I would like to see empirical evidence on how many flips occur during the experiments. If no flips are observed, the current mechanism is likely sufficient to avoid flips. W6. The number of vertices is never mentioned. There is a comment that a 30^3 tetrahedral grid is equivalent to a 70^3 voxel grid, but more discussion is needed on how many vertices are used for the results shown and what is the maximum number that can be handled with the current implementation.

Correctness: The paper appears to be correct. The limitations of the approach should be stated more clearly (see weaknesses above). The smoothness loss is described imprecisely, but it can be corrected.

Clarity: The paper is well written, but too much information is compressed to fit within the page limit. The supplement is essential for understanding the loss function and the network design at a coarse level. The diversity of applications is part of the appeal of DefTet, but describing one application fully in the main paper and the rest in the supplement may have been a better approach.

Relation to Prior Work: The differences with previous learning approaches are described clearly. There are, however, missing references from the conventional 3D reconstruction literature, the most important of which are from Keriven et al.: Patrick Labatut, J-P Pons, and Renaud Keriven. Robust and efficient surface reconstruction from range data. Computer Graphics Forum, 28(8):2275-2290, 2009. H.-H. Vu, P. Labatut, J.-P. Pons, and R. Keriven. High accuracy and visibility-consistent dense multiview stereo. IEEE Transactions on Pattern Analysis and Machine Intelligence, 34(5):889{901, May 2012.

Reproducibility: Yes

Additional Feedback: I only have a few minor comments. The list below is not necessarily exhaustive. Both “tetrahedralize” and “tetrahedronize” are used. The former should be preferred. 289: “to quantify the performance of different methods” 300: the numbers of failures are inconsistent in the text and Table 3. [17]: missing publication venue. POST REBUTTAL COMMENTS The rebuttal was carefully prepared, but some of the limitations were correctly pointed out by the reviewers. I see this as an imperfect method, but the fact that it proposes a very interesting and potentially powerful representation outweighs the limitations, in my opinion. As long as the limitations are stated, I think that the paper can be accepted.


Review 4

Summary and Contributions: This paper proposes a new data representation for 3D object reconstruction: Deformable Tetrahedral Mesh (DefTet). DefTet is formulated as deformed regular voxel grid by per-point displacement vectors. Along with per-tetrahedra opacity, the representation supports arbitrary topology (compared to regular mesh) and flexible geometry (compared to regular voxel). Further it does not require post-processing as in implicit function. The authors demonstrate the performance of the proposed representation in terms of point cloud 3D reconstruction, single-view image 3D reconstruction (supervised or self-supervised by differentiable rendering similar to NeRF or Differentiable Ray Consistency [Tulsiani et al CVPR 17]) and tetrahedral meshing. The proposed representation shows promising results in reconstruction quality and relative advantage on speed (compared to implicit surface), on representation efficiency (compared to regular voxel) and on topology flexibility (compared to regular mesh).

Strengths: I think the representation of DefTet is a simple but rather effective design. It hits the sweet spot among the existing representations: on speed (compared to implicit surface), on representation efficiency (compared to regular voxel) and on topology flexibility (compared to regular mesh). The Tetrahedral mesh was previously common in physical simulation, therefore this new design might help in learning physics. The presentation and writing of this paper is good.

Weaknesses: 1. Conceptually I think the closest work to this new representation DefTet is Deep Marching Cube (DMC) [26]. In their work, they also predict an occupancy grid and displacement grid, and based on that computes a surface. DMC is potentially also capable of handling various topologies and the computation seems at similar level with DefTet. Therefore I consider it important to have a proper comparison and discussion on DMC. Some experimental comparison would also be good (similar to Table 1, but for other experiment categories). More specifically, if the displacement in DefTet is not too large (regularized by the L_del and L_lap), the formulation is very similar to DMC (they constraint the displaced vertices on [0, 1]), except for their displaced vertices only live on the grid edges instead of freely. How much does the displacements of DefTet typically be? Is that flexibility the reason that DefTet performs better? Here I’m still trying to understand the difference and an explanation for the difference in performance. Also DefTet is significantly faster than DMC in inference time (8x-10x faster). Any good explanation? 2. Would be good to show ablation studies on the loss functions. The loss function contains many terms, it’d be good to see their necessity and individual contribution. 3. What would be the drawback of Laplacian smoothing as the last layer of the whole differentiable computational graph? (as mentioned in line 219 in section 4) 4. The comparison to the traditional tetrahedral meshing method does not give a very clear conclusion. The proposed method indeed is robust, but still quite inaccurate compared to TetGen and QuarTet, even without learning (direct optimization), at the cost of failing rate below 1%. Also, in line 300 it says 40 failed on TetGen and 3 failed on QuarTet. This does not align with table 3 (typo?)

Correctness: Yes, except for those points listed in the weakness session, which a rebuttal response is expected.

Clarity: Yes

Relation to Prior Work: As mentioned in the weakness session, a proper comparison and discussion on Deep Marching Cube should be added.

Reproducibility: Yes

Additional Feedback: After rebuttal: Thanks for the rebuttal. Most of my concerns about DMC are addressed. I would encourage the authors to add more discussion / comparison with DMC. Please also add sufficient ablation studies on loss functions and comparisons as mentioned in the review comments, and fix typos and mistakes. Other comments: For your response "DMC is not applicable to single image 3D reconstruction using 2D supervision", to my understanding, adding another mesh-based differentiable rendering after the DMC would make self-supervised training work?

[Author Response · NeurIPS 2020]



Figure 1: Ablation studies on regularization terms. We sequentially add regularizations from left to right. We show the corresponding Chamfer Distance↓ (CD) & AMIPS↓ on the bottom.

GT · $L_{\text{recons}}$ 3.531&8.20 · $+L_{\text{lap}}$ 3.327& 7.13 · $+L_{\text{del}}$ 3.325 & 6.80 · $+L_{\text{vol}}$ 3.329& 4.76 · $+ L_{\text{amips}}$ 3.336& 3.62 · $+ L_{\text{sm}}$ 3.330 & 3.64

Figure 2: Scene.

We would like to thank reviewers for their detailed comments. We address questions below.

**[R1,R2,R3,R4] Ablation Studies:** We provide a qualitative ablation study for different terms by optimizing DEFTET
to ground truth shape in Fig. 1. $L_{\text{lap}}$ helps maintain the regularity. $L_{\text{del}}$ encourages local deformation. $L_{\text{vol}}$ helps avoid
flips, $L_{\text{amips}}$ helps avoid needle-like predictions, and $L_{\text{sm}}$ helps smooth the prediction. We will add these in a revision.

**R1 Impact of Resolution:** We ablated different resolutions of DEFTET compared to a voxel-based representation in
Fig. 4. We report the performance of DEFTET at 70 resolution in Table 1 compared to SOTA SDF-based approach [40].

**R1 More convincing application:** We partly disagree. Several algorithms have been proposed to differentially render
an implicit function, yet these methods still use the non-differentiable marching cube algorithm to convert SDF into a 3D
mesh, which impedes inference speed. DEFTET predicts meshed shape without the time-consuming post-processing.

**R2 Unfair comparison to Pixel2Mesh [46] and MeshRCNN [14]:** $L_{\text{vol}}$ and $L_{\text{amips}}$ are volume-based and not suitable
for [46]&[14]. $L_{\text{lap}}$ is also used in [46]&[14]. $L_{\text{sm}}$ and $L_{\text{del}}$ can also be used in [46]&[14]. We perform an additional
experiment on point cloud 3D reconstruction, [46]/[14] achieved 66.06/72.92 IOU, even worse than original results.

**R2 No comparison with SOTA SDF approaches:** We wish to clarify that we did compare with SOTA SDF-based
approaches. For point cloud 3D reconstruction (Tbl. 1, Fig. 3&4), we compared with [40] (ECCV2020), the best
SDF-based approach at the time of our submission. For single image 3D reconstruction with **2D supervision** (Tbl. 2,
Fig. 5), where **no** 3D ground truth is utilized, we compared with DVR [37], which is the SOTA SDF-based approach on
this task. In both tasks, we outperformed the baselines. DISN is not applicable in these two tasks as it is specifically
designed for single image 3D reconstruction using **3D supervision**, i.e., 3D ground truth is required during training.

**R2 Worse result on single image 3D reconstruction:** We first clarify that single image 3d reconstruction can be
done using 3D or 2D ground truth as supervision, and we apply DEFTET in both regimes. We show results of
using 2D supervision in the main paper, which has significant improvement, and we report **both quantitative and
qualitative** results of using **3D supervision** in the supplement (Tbl. 2, Fig. 3). Due to the deadline rush, the results
in supplement were reported with a non converged network – we apologize. We report the true performance as
2.006/19.463 Chamfer/Hausdorff Mean, which outperforms [46]&[14]. We thank the reviewer for pointing out DISN.
Since we use the network backbone from OccNet, it is not fair to directly compare with DISN which uses a more
powerful network structure. We will include a fair comparison in the revision.

**R2 Concerns on Eq. 9∼11:** We tried both and found that considering all faces converges faster than only considering
the last face. If $f\_1$ is partially occluded by $f\_2$, the occluded part would have a much lower visibility due to Eq.10
($m_1 < m_2$). Note that we use $m$ instead of $D$ to accumulate colors.

**R3 Experiments on other categories:** We showed cars in the supplement, and we will add more classes in a revision.

**R3 #Vertices:** This indeed is a limitation, as is for the voxel-based approaches, yet less so for DEFTET since we can
deform the grid geometry. In the future, we plane to adaptively subdivide to get details, akin to oct-tree voxel grids.

**R3 Scale to large scene:** We partially agree with the reviewer, but we can increase the resolution of DEFTET for the
complex scene. We show two optimization results in Fig. 2 by reconstructing multiple objects with DEFTET.

**R3 Smoothness term:** $\mathcal{F}$ here denotes the set of all triangular faces whose $P_s(v)$ equals to one (L185 in the paper),
instead of all the faces of the tetrahedralization. We want the neighbouring boundary faces to be approximately parallel.
We apologize for the typo – there should be a negative sign before cosine. We will revise.

**R3 Avoid flips:** We thank the reviewer for raising this question. There is a tradeoff between having no flips and
performance. We can achieve having no flips by increasing the weight of $L_{\text{vol}}$ but sacrifice the performance. Empirically,
when doing optimization, we achieve 3.783 CD with 0 flip, and 3.330 CD with 1 flip over all 729162 tetrahedrons.

**R3 #Vertices:** The #vertices for $30^3$ is 4198, and $70^3$ is 47416 , the maximum resolution we can support is $150^3$ with
batch size 4 on a 32G V100 GPU. **R3 Related literature:** Thank you for pointing this out, we will add it.

**R4 Comparison with DMC [26]:** We agree that we share similarities with DMC – we discussed (L90) and compared
with (Tbl. 1 & Fig. 3) DMC. We differ in the following aspects: **1. Basic element**. DMC deforms vertices in a cube
and constraints the deformation to be along the edges within [0,1], while we freely deform tetrahedron's vertices,
whose typical displacements are ∼ 1.18x of edge length. Therefore, DMC has a fixed set of possible topologies and
needs to deal with many corner cases (Fig. 2 in [26]), while we have more flexibility in representing geometric details
(e.g. the curvy chair back in Fig. 3) and no corner cases. **2. Formulation.** DMC focuses on the surface and uses the
"occupancy" to derive the probability over possible topologies. While our DEFTET explicitly defines the occupancy
for one tetrahedron and deforms it which makes training easier than in DMC. We compared with DMC in point cloud
3D reconstruction task and single image 3D reconstruction using 3D supervision, and achieved significantly better
performance. DMC is not applicable to single image 3D reconstruction using 2D supervision. The reason why DMC
runs slower is that the official source code of DMC runs an iterative algorithm to get the mesh after obtaining the
displacement and occupancy. We refer to the source code for details.

**R4 Drawback of laplacian smoothing layer:** We do not notice drawbacks. The computation is neglible.

**R4 Comparison with Tet Meshing:** We show that our DEFTET reaches comparable performance to traditional
tetrahedral meshing methods – a sanity check experiment. We are more accurate than QuarTet in Tbl. 3. The robustness
(even 1% failing rate) are very important for applications in computer graphics [17]. Note that all of traditional methods
can not be plugged into deep learning, while DEFTET can easily support it. We apologize for the typo, Tbl. 3 is correct.

[Meta-Review · NeurIPS 2020]

Consensus among the reviewers was to accept this paper. In your final revision, please address comments in the reviews and any promises in the rebuttal. Specifically, comprehensive comparison to DMC and ablation studies are needed to sure up confidence in the method and its relationship to past work.